# Evaluation of the Reporting Standard Guidelines of Network Meta-Analyses in Physical Therapy: A Systematic Review

**DOI:** 10.3390/healthcare10122371

**Published:** 2022-11-25

**Authors:** Sung-Hyoun Cho, In-Soo Shin

**Affiliations:** 1Department of Physical Therapy, Nambu University, 23 Cheomdan Jungang-ro, Gwangsan-gu, Gwangju 62271, Republic of Korea; 2AI Convergence Education, Graduate School of Education, Dongguk University, 30, Pildong-ro 1 gil, Jung-gu, Seoul 04620, Republic of Korea

**Keywords:** systematic review, network meta-analysis, treatment outcome, checklist, randomized controlled trials

## Abstract

The concept of network meta-analyses (NMA) has been introduced to the field of physical therapy. However, the reporting standard guidelines of these studies have not been evaluated. In this systematic review, we included all published NMA physical therapy studies that compared the clinical efficacy of three or more interventions to evaluate whether NMAs in physical therapy exhibit adequate reporting recommendations. PubMed, EMBASE, Web of Science, and the Cochrane Library were searched up to 30 June 2022. Among the 252 identified articles, 19 NMAs including 805 randomized controlled trials were included. We applied both preferred reporting items for systematic reviews and meta-analysis (PRISMA) and PRISMA-NMA checklists, which are 27- and 32-item reporting standard guidelines assessment tools, respectively. Protocol registrations (68.4%), risk of bias across studies (63.2%), additional analysis (57.9%), and funding (31.6%) were problematic items considering the PRISMA guidelines. Four studies reported all five new NMA-reporting items, and 15 (78.9%) did not address items S1–5 from the PRISMA-NMA guidelines. The median score (interquartile range) of the reporting standard guidelines was 27.0 (25.8–28.0). The identified shortcomings of published NMAs should be addressed while training researchers, and they should be encouraged to apply PRISMA-NMA, as a recognized tool for assessing NMA reporting guidelines is required.

## 1. Introduction

Physiotherapists are becoming accustomed to the use of research findings in evidence-based practice. Physiotherapists interested in using research findings to determine the optimal physiotherapeutic intervention for a patient may look for high-quality systematic reviews and meta-analyses [1]. Assessing the comparative effectiveness of many or all available interventions for clinical indications is challenging [2]. In the hierarchy of evidence, a network meta-analysis (NMA) synthesizing several different treatment effects from randomized controlled trials (RCTs) comparing two different treatments is considered as the superior treatment-effect meta-analysis [3]. However, few studies on the methodological quality of NMAs have indicated significant shortcomings, especially in terms of the statistical methodology and analytical process in clinical research articles [4,5,6].

Various physiotherapeutic intervention techniques exist; however, there are no practical guidelines to assist clinicians in choosing the best technique for individual patients [7,8]. When systematically reviewing the literature, directly comparing the differences between treatments of interest is preferable [9]. However, many reviews have not directly compared the treatments [10]. Recently, the concept of NMA has been introduced to the field of physical therapy [1]. Reviews of the published applications of NMAs and indirect comparisons have revealed that assumptions are often not assessed. When an assumption is assessed, the reporting of the methods and results may be insufficient, and the methods applied may not be consistent across reviews [11]. Most of these reviews have focused on assessing the methodological quality of NMAs, especially the validity assumptions (based on heterogeneity, transitivity, and inconsistency), which are considered important [12]. Different studies have examined and evaluated the reporting standard guidelines of NMAs such as evaluation papers of reporting standard guidelines for other intervention meta-analyses in the pharmacological intervention [13], Chinese medicine [14], and complementary and alternative medicines [15]. However, conducting a NMA has several challenges and limitations in terms of the indication of an existing review protocol, exploration of network geometry, and risk of bias across studies such as publication bias in complementary and alternative medicines [15]. In recent years, NMA reviews have been gradually increasing in the field of physical therapy; however, no one has evaluated whether these reviews were performed with adequate evaluation of the reporting standard guidelines.

Therefore, we conducted this systematic review of published NMAs to examine whether the reports adequately followed the key reporting components of the systematic review process, based on the preferred reporting items for systematic reviews and meta-analysis (PRISMA)-NMA Extension guidelines, which were recently developed based on the consensus of NMA experts [16].

Despite the increase in NMA research in physical therapy, none of these were evaluation papers of the reporting guidelines. Therefore, this much-needed study aimed to answer the following question: Do NMAs in physical therapy exhibit adequate reporting guidelines?

## 2. Materials and Methods

### 2.1. Registration

Our study protocol was registered with the International Prospective Register of Systematic Reviews (PROSPERO); registration number: CRD42020209965.

### 2.2. Information Sources and Search Strategy

We assessed the data in accordance with the PRISMA guidelines for NMA [16]. All NMAs published in physical therapy journals comparing the clinical efficacy of three or more interventions based on RCTs were considered eligible. We only included NMAs with at least one physical therapy in the set of treatments examined and the data from at least three clinical trials. We also included NMAs that compared the efficacy of the treatments. We excluded methodological reviews, editorial style reviews or concise reviews, conventional meta-analysis reviews, articles not related to physical therapy journals, and reviews not involving human participants.

Two reviewers independently reviewed the titles and abstracts of all studies retrieved by the search. Duplicates were removed using Endnote X9 (Thomson Reuters Co., New York, NY, USA). Full-text articles were obtained and examined, if necessary. The reviewers then selected potentially relevant studies according to the eligibility criteria (Table 1).

The search strategy was a combination of medical subject heading [MeSH] terms and free text words including “network meta-analysis” [MeSH], “review” [MeSH], “systematic review” [txt words], “physical therapy” [MeSH] AND “physiotherapy” [txt word], “Physical therapy modalities” [MeSH], “Physical Therapy Specialty” [MeSH], “Physical Therapy Modalities” [MeSH].

The keywords, with the MeSH terms of the search strategies for all electronic database searches, are listed in Table 2. We limited our search to studies including RCTs with human participants and we only included articles published in English-language journals.

### 2.3. Identification of Studies

We comprehensively screened eligible studies on the efficacy of physical therapy from the EMBASE (http://www.embase.com, accessed on 15 July 2022), PubMed (http://www.ncbi.nlm.nih.gov/pubmed/, accessed on 15 July 2022), Web of Science (http://webofknowledge.com/, accessed on 15 July 2022), and Cochrane Library (http://www.cochranelibrary.com, accessed on 15 July 2022) databases, published up to 30 June 2022. The search terms used in EMBASE, PubMed, Web of Science, and the Cochrane Library are presented as examples in Appendix A [7,8,10,17,18,19,20,21,22,23,24,25,26,27,28,29,30,31,32]. 

### 2.4. Hand Search

Reference lists of the included and previously published systematic reviews with NMA related to the topic were screened to identify any additional studies.

### 2.5. Study Selection

Each identified article was independently screened by title and abstract by the authors to remove duplicate entries and studies that failed to meet the inclusion criteria. To avoid excluding potentially relevant articles, the full-text paper was searched and examined when the abstract provided unclear information. Any disagreements were resolved through discussion. Full-text articles that satisfied the inclusion criteria were assessed by two reviewers with clinical knowledge of physical therapy and methodological knowledge of NMA. The references of the included articles were further checked manually.

Two trained reviewers independently extracted data from each included study and resolved disagreements through consensus. We reviewed all of the published materials related to each NMA including online supplementary materials and appendices.

### 2.6. Data Collection Process and Coding Items of Clinical Study Characteristics

The titles and abstracts of all articles identified by the search strategy were evaluated in duplicate by two independent investigators. All abstracts that did not provide sufficient information regarding the eligibility criteria were selected for full-text evaluation. In the second phase, the same reviewers independently evaluated the full-text articles and made their selections in accordance with the eligibility criteria. Disagreements between the reviewers were resolved through discussion. The authors independently extracted the data from each included article into predesigned coding sheets: (1) Study identification: first authors’ name, location of corresponding authors, year of publication, and journal name; (2) number and design of studies included in the NMA; (3) population (participants); (4) interventions; (5) comparison between interventions; and (6) outcome measures. Detailed information is provided in Appendix A.

### 2.7. Reporting of General Components and Key Methodological Components of the Systematic Review Process

This methodological systematic review was conducted in accordance with the recommendations of the PRISMA Extension guidelines for reporting NMAs [16], which includes a 32-item checklist and flow diagram: 26 general items and five new NMA items as well as 11 modifications to previous PRISMA items for the 27 general items. To support valid and reliable clinical decision-making, we assessed whether key methodological and general components were reported.

### 2.8. Evaluation of Reporting Standard Guidelines Assessment Tools

We also applied both PRISMA and PRISMA-NMA checklists (http://www.prisma-statement.org/, accessed on 15 July 2022), which are 27- and 32-item tools, respectively, identifying the relevant information that should be reported by authors in a systematic review with pairwise or NMA. “The PRISMA Extension Statement for Reporting of Systematic Reviews Incorporating Network Meta-analyses of Health Care Interventions: Checklist and Explanations” (PRISMA-NMA) [16], which comprises 32 items, was used by the two independent authors to evaluate the reporting standard guidelines. For articles published after June 2015, the PRISMA-NMA was considered the standard checklist. For articles published from July 2009 to June 2015, the PRISMA checklist was used.

Each item was scored as “1” for full compliance, “0.5” for partial compliance, and “0” for non-compliance [33,34]. The summary PRISMA-NMA score for a NMA was calculated by accumulating the scores of each item, with a possible maximum of 32. To quantitatively assess the results, after excluding non-applicable items of the tools (e.g., given results of additional analyses when no additional analyses were done), we applied a “yes” (1 point) and “no” (0 points) scale to each item, creating a maximum score of 27 and 32 points for the PRISMA and PRISMA-NMA, respectively. A score of 26 to 32 was identified as “high”, 20 to 25.5 as “moderate”, and 19.5 or lower as “low” [14].

### 2.9. Statistical Analysis: Descriptive Statistics and Frequency

Categorical data were summarized as number (percentage). The extracted information on the PRISMA items was summarized across the eligible systematic reviews using absolute and relative frequencies. Statistical analyses were performed using SPSS version 26.0 (IBM Corp., Armonk, NY, USA). This study mainly used descriptive statistics and frequency based on the standard reporting guidelines.

### 2.10. Ethical Approval

This systematic review did not require ethical approval considering that only retrospective literature was included and evaluated.

## 3. Results

### 3.1. Study Selection

We identified 252 publications (Figure 1) through electronic database searches. After eliminating duplicates, 129 articles were selected, 74 of which were excluded after screening the title and abstract. We reviewed 55 articles for eligibility by assessing the full text. The reasons for study exclusion during the final review were as follows: editorial style review (*n* = 2), systematic review (*n* = 2), study protocol systematic review (*n* = 5), lack of complete data (*n* = 4), articles not involving criterion of intervention (*n* = 20), and articles that were not RCTs (*n* = 3). We included the remaining 19 articles to evaluate the reporting standard guidelines of NMAs for physical therapy (Figure 1).

### 3.2. Study Characteristics

#### 3.2.1. Epidemiological and Descriptive Characteristics

The 19 included NMAs were published in 16 journals: three reports were published in the British Journal of Sports Medicine, and two in the Archives of Physical Medicine and Rehabilitation (Appendix A). The corresponding authors were in Asia (11, 57.9%), Europe (4, 21.1%), Oceania (3, 15.8%), and South America (1, 5.2%). The types of interventions varied across wide-ranging physical therapy fields. The form of intervention was largely divided into different combined exercise training (73.7%) and physical therapy modalities (26.3%) (Appendix A). The reporting guidelines used for the systematic review process also varied: eight (42.1%) articles used the PRISMA guidelines (2009) and five (26.3%) used the PRISMA-NMA Extension reporting guidelines (2015). However, six (31.5%) studies did not describe the reporting guidelines that were used. Regarding the funding sources, six studies received private and/or public support (31.5%), and 13 studies did not clearly report their funding source.

#### 3.2.2. Clinical and General Characteristics of Included Studies

The main characteristics of all the included studies are described in Table 3. The included studies were published between 2013 and 2020. Based on these results, we determined that most studies were conducted by a group of researchers. Among the 19 NMA studies, 10 used the Bayesian statistical approach, and the remaining nine used the frequentist NMA approach. Furthermore, 18 (94.7%) studies included RCTs, while the remaining one (5.3%) included both RCTs and non-RCTs. For the comparison of three or more interventions, most NMAs used all possible comparisons among the different interventions, and two did not use a placebo (control) group as a common comparator [25,26]. The ability of NMAs to incorporate indirect evidence means that the inclusion of interventions that are not of direct interest to the review authors may provide additional information in the network. For example, a placebo is often included in NMAs, even though it is not a reasonable treatment option, because many studies have compared active interventions against placebos. In such cases, the exclusion of a placebo would have resulted in ignoring a considerable amount of indirect evidence [9,35]. A common comparator is important for estimating the indirect effects. Most studies used the outcome of continuous variables such as the differences in the mean change in adults with hemodialysis, plantar fasciitis, mild cognitive impairment, lateral mechanical ankle instability, knee or hip osteoarthritis, overweight and obesity, non-specific chronic low back pain, chronic calcific tendinitis of the shoulder, and patients with post-stroke dysphagia, type-2 diabetes mellitus, carpal tunnel syndrome, Parkinson’s disease, and cancer.

## 4. Reporting Standard Guidelines Assessment

After assessing the compliance of the NMAs using the 32-item PRISMA-NMA checklist, we obtained a median (interquartile range) score of 27.0 (25.8–28.0), but no NMA met all 32 items, with the full details provided in Table 4.

### 4.1. Reporting of Key Methodological Components of the NMA Process (New Items, S1–5)

The report of the key methodological components of the NMAs is presented in Table 4. The transitivity assumption refers more to the conceptual homogeneity of the study design, participants, and methodologies, whereas consistency denotes more the statistical homogeneity of parameters and variables of the trials of different comparisons [36]. In the Methods section, 13 (68.4%) articles did not describe the methods used to explore the geometry of the treatment network (S1). Furthermore, the assessment of inconsistency such as differences in the statistical methods used to evaluate the agreement of direct and indirect evidence were not reported in four (21.1%) studies (S2). Finally, four (21.1%) studies did not include investigations of inconsistency (S5).

### 4.2. Descriptive Analysis

Descriptive statistics were summarized as publications that reported each item of the PRISMA-NMA (Table 5). On setting the impact factor (high or low) criterion to a median of 3.607, the high impact factor group comprised a median of 27.5, and the low impact group comprised a median of 25.6; however, the difference was not statistically significant (Table 6).

## 5. Discussion

The key reporting components of the systematic review process were missing in most NMAs.

### 5.1. Reporting of 16 Items from the Original PRISMA Statement

In the Methods section, only six (31.6%) NMAs reported the protocol registered in PROSPERO (item 5). Protocol registration is a procedure to prevent selective reporting and to make more valid and consistent results for clinicians. In addition, six (31.6%) studies did not list and define all variables for which data were sought (for example, PICOS, and funding sources) and any assumptions and simplifications made (item 11). Furthermore, two studies were published before 2015 and 17 were published after 2015. However, only five papers were PRISMA-NMA-endorsing (item 11). Three papers applied the Grading of Recommendations Assessment, Development, and Evaluation (GRADE) approach in quality evaluation [17,21,30]. The GRADE approach was used to assess the quality of the evidence behind the ranking of treatments from NMAs [37] (item 12).

The assessment tool, Risk of Bias 2 (ROB2) (Cochrane risk of bias scale), was revised in June 2019. In future meta-analysis studies, risk of bias can be meaningfully interpreted using the ROB2 tool/ROB in Non-randomized Studies (ROBINS) tool (item 12). Furthermore, six (31.6%) NMAs did not report the methods used to assess the risk of bias across studies (for example, publication bias) (item 15). Publication bias items were not reported in seven studies (item 22).

### 5.2. Reporting of 11 Modified Items

Seventeen NMA abstracts did not describe the NMA assumption in detail (item 2). In the Methods section, 18 papers did not clearly describe eligible treatments included in the treatment network and did not report whether the treatments were clustered or merged into the same node with justification (item 6). Lumping of interventions (cluster or merge) requires treatments with similar treatment effects. Although this technique is appropriate in some cases, it should be clearly rationalized when performed.

All 19 studies used the standardized mean difference as a summary measure. Ranking probability was reported using rankograms (8/19, 42.1%) and the surface under the cumulative ranking curve (10/19, 52.6%). League tables (17/19, 89.5%) and forest plots (15/19, 78.9%) were used, and seven studies reported using effect size, variance formula, and weight (36.8%) (item 13). Bayesian model fit was evaluated using the deviance information criteria (DIC) (5/10, 50%) [38]. The program for performing NMAs consisted of eight studies using the R software and 11 using the Stata software (item 14). In the results section, seven (36.8%) did not report the risk of bias assessment across the included studies (item 22). The alternative network geometries and alternative choice of prior distributions for Bayesian analyses were mentioned in only 11 (57.9%) NMAs (item 23). In the Discussion section, comments on the validity of assumptions such as transitivity and consistency, were not reported in seven (23%) NMAs (item 25).

### 5.3. Reporting of Five New Items and NMA Assumptions

On reviewing the network geometry (item S1), 13 (68.4%) studies did not report this in the Methods section. A qualitative description of the network geometry should be provided, accompanied by a network graph [16,39]. The statistical test of inconsistency was assessed by inspecting the available evidence (78.9%) (item S2). To evaluate the consistency assumption, 11 studies (52.6%) used a global approach and 13 (68.4%) used a local approach. Local approaches assess the presence of inconsistency for particular pairwise comparisons in the network, whereas global approaches consider the potential for inconsistency in the network as a whole [40]. Thus, evaluating the consistency assumption using both global and local approaches is generally recommended [16]. DIC can be used to consider the model fit (five studies, 50.0%) [40]. The global inconsistency approach and DIC index are not well reported because these items are not familiar to clinical researchers.

The exploration for the inconsistency area (item S5) revealed 15 studies (78.9%). The global, local, and both model fits were reported in the Results section of 10 (52.6%), 13 (68.4%), and eight (42.1%) papers, respectively. Both global model-fit and local model-fit should report well, but most NMA studies do not report well. Eleven papers (58%) did not report that both models fit well in the Results section. These NMAs had some limitations due to the characteristics of the included studies. First, most trials included in the analyses had uncertain or high risks of bias. Second, the efficacy of some interventions might have resulted from inadequate estimates of inconsistency due to a non-closed loop. Third, the PRISMA-NMA Extension guidelines are relatively new (2015); hence, their adoption may take 2 or 3 years for busy clinicians.

### 5.4. Limitation and Implication of the Study

To our knowledge, no study has investigated the systematic review process of NMAs in physical therapy. The score of reporting standard guidelines in this paper is the summation of PRISMA-NMA 32 items, a kind of frequency score. This score of reporting standard guidelines in this paper had some limitations because each item of the PRISMA guideline has its own meaning. The limitation of this study is the interpretation of the score for reporting the standard guidelines. The score for the reporting standard guidelines, which is the summation of 32 items, should be cautiously interpreted. The most important research method for establishing an evidence-based practice system is a systematic review. A systematic review can be used as supporting data by systematically arranging research data, and reproducible objectivity is secured regardless of the researcher. This study will be helpful in nurturing clinical experts by providing data for evidence-based clinical decision-making in the field of physical therapy. It can be used as basic data that can contribute to the establishment of efficient and systematic medical policies. Improving the quality of evidence for patient physical therapy programs will contribute to policy development related to the patients’ physical therapy and to the improvement in national health medical services by improving the treatment quality of clinical practitioners.

## 6. Conclusions

This critical assessment demonstrated that the current evaluation of reporting standard guidelines and conducting NMAs is low to moderate as in other medical disciplines. The focus areas identified in the current NMAs include exploring the network geometry, the assessment of inconsistency, risk of bias across studies, protocol registrations, and additional analyses. NMAs with inadequate reporting increase the risk of producing invalid results. The identified shortcomings of the published NMAs should be taken into consideration in the further training of authors and editors of NMAs in physical therapy. Therefore, reporting guidelines such as the PRISMA Extension Statement are helpful for authors when reporting NMAs. Moreover, researchers should be encouraged to apply the PRISMA-NMA. Although the PRISMA Extension guidelines are relatively new, they should be used more extensively and endorsed in journal author guidelines to improve the reporting practices in physical therapy.

## Figures and Tables

**Figure 1 healthcare-10-02371-f001:**
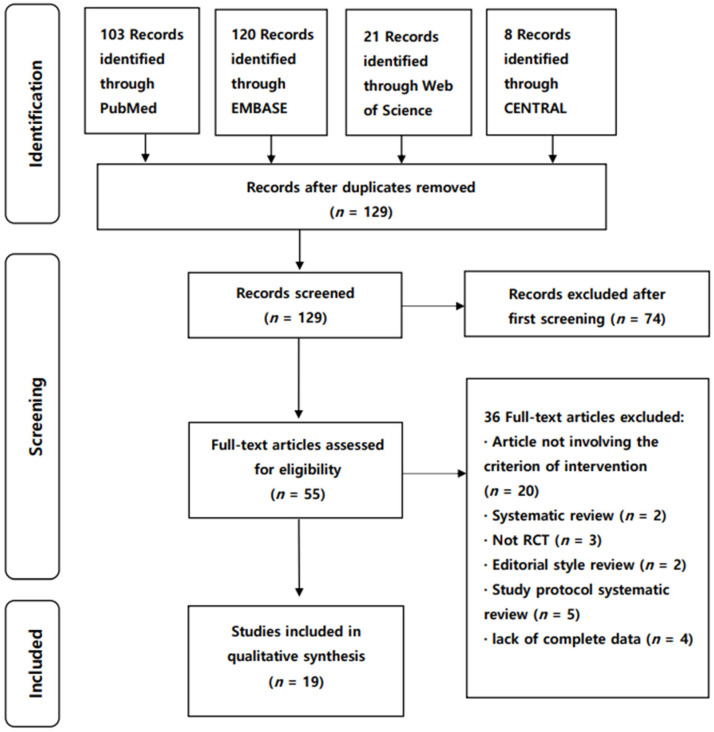
PRISMA flow diagram of the included NMAs. NMA, network meta-analyses; RCT, randomized controlled trial.

**Table 1 healthcare-10-02371-t001:** Inclusion criteria.

Criteria	Determinants
Population	Adults or patients related to physical therapy
Intervention	At least one group received physical therapy
Comparison	At least one group received a comparison intervention or no intervention
Outcome measures	Functional outcome relating to physical therapy
Design	Randomized trial with physical therapy as treatment

**Table 2 healthcare-10-02371-t002:** Detailed search strategies for each database.

Database	Detailed Search Strategies	Records Found
PubMed	(“network meta-analysis” [MeSH] OR “network meta-analyses” [all] OR “network meta analysis” [all] OR “network meta analyses” [all]) AND (“Physical Therapy Modalities” [MeSH])	73
	(“network meta-analysis” [MeSH] OR “network meta-analyses” [all] OR “network meta analysis” [all] OR “network meta analyses” [all]) AND (“Physiotherapy” [all])	30
	(“network meta-analysis” [MeSH] OR “network meta-analyses” [all] OR “network meta analysis” [all] OR “network meta analyses” [all]) AND (“Physical Therapy Specialty” [MeSH])	0
EMBASE	(‘network meta-analysis’/exp OR ‘network meta-analysis’ OR ‘network meta-analyses’/exp OR ‘network meta-analyses’ OR ‘network meta analysis’/exp OR ‘network meta analysis’ OR ‘network metaanalysis’/exp OR ‘network metaanalysis’) AND (‘physical therapy modalities’/exp OR ‘physical therapy modalities’)	31
	(‘network meta-analysis’/exp OR ‘network meta-analysis’ OR ‘network meta-analyses’/exp OR ‘network meta-analyses’ OR ‘network meta analysis’/exp OR ‘network meta analysis’ OR ‘network metaanalysis’/exp OR ‘network metaanalysis’) AND (‘physiotherapy’/exp OR ‘physiotherapy’)	58
	(‘network meta-analysis’/exp OR ‘network meta-analysis’ OR ‘network meta-analyses’/exp OR ‘network meta-analyses’ OR ‘network meta analysis’/exp OR ‘network meta analysis’ OR ‘network metaanalysis’/exp OR ‘network metaanalysis’) AND (‘physical therapy specialty’/exp OR ‘physical therapy specialty’)	31
Web of Science	TOPIC: (“network meta-analysis” OR “network metaanalysis”) AND (“Physical Therapy Modalities” OR “Physical Therapy Specialty” OR “Physiotherapy”)	21
Cochrane Central Register of Controlled Trials	(network meta-analysis OR network metaanalysis) AND (Physical Therapy Modalities OR Physical Therapy Specialty OR Physiotherapy)	8

Medical subject heading (MeSH) terms, search terms, and combinations of the two were used for each database search. Ultimately, 252 records were identified: 103 from PubMed, 120 from EMBASE, 21 from the Web of Science, and eight from the Cochrane Library. Studies were further selected according to the inclusion criteria.

**Table 3 healthcare-10-02371-t003:** Comparison of the positive results obtained by NMAs using the PRISMA and PRISMA-NMA checklists.

**No.**	**Items**	**Checklist (Summary)**	**PRISMA** **n (%)**	**PRISMA-NMA** **n (%)**	
	**Common items for PRISMA and PRISMA-NMA**				
4	Introduction, objectives	Provide an explicit statement of questions (e.g., PICOS)	18 (94.7%)	18 (94.7%)	
5	Methods, protocol and registration	Indicate whether a review protocol was followed, provide the registration number	13 (68.4%)	13 (68.4%)	
7	Methods, information sources	Describe all information sources in the search and date last searched	19 (100%)	19 (100%)	
8	Methods, search	Present the full electronic search strategy for at least one database	19 (100%)	19 (100%)	
9	Methods, study selection	State the process for selecting studies	18 (94.7%)	18 (94.7%)	
10	Methods, data collection process	Describe the method of data extraction from the selected reports	19 (100%)	19 (100%)	
11	Methods, data items	List and define all variables for which data were sought	13 (68.4%)	13 (68.4%)	
12	Methods, risk of bias within studies	Describe the methods used to assess the risk of bias	18 (94.7%)	18 (94.7%)	
15	Methods, risk of bias across studies	Specify any risk of bias assessment that might have affected the cumulative evidence	13 (68.4%)	13 (68.4%)	
17	Results, study selection	Provide the numbers of studies screened, assessed for eligibility, and included	19 (100%)	19 (100%)	
18	Results, study characteristics	For each study, present the characteristics for which data were extracted	19 (100%)	19 (100%)	
19	Results, risk of bias within studies	Present data on the risk of bias of each study	19 (100%)	19 (100%)	
22	Results, risk of bias across studies	Present the results of assessment of risk of bias across studies	12 (63.2%)	12 (63.2%)	
24	Discussion, summary evidence	Summarize the main findings including the strength of evidence	19 (100%)	19 (100%)	
26	Discussion, conclusion	Provide a general interpretation of the results and their implications	19 (100%)	19 (100%)	
27	Funding	Describe sources of funding	6 (31.6%)	6 (31.6%)	
	**PRISMA items modified for PRISMA-NMA**				
1	Title	Identify the report as a systematic review incorporating a network meta-analysis (or related form)	19 (100%)	17 (89.5%)	
2	Abstract, structured summary	Provide a structured summary mentioning the network meta-analysis	19 (100%)	19 (100%)	
3	Introduction, rationale	Describe the rationale for the review; mention why a network meta-analysis has been conducted	19 (100%)	19 (100%)	
6	Methods, eligibility criteria	Specify study characteristics, clearly describe eligible treatments included in the network	11 (57.9%)	1 (5.3%)	
13	Methods, summary measures	State the principal summary measures, describe the use of additional measures such as treatment rankings, SUCRA values	19 (100%)	16 (84.2%)	
14	Methods, planned analyses	Describe the methods of handling data and combining results (e.g., handling of multi-arm trials; selection of variance structure; prior distributions in Bayesian analyses; model fit)	19 (100%)	9 (47.4%)	
16	Methods, additional analyses	Describe the methods of additional analyses (e.g., sensitivity or subgroup analyses, meta-regression, or alternative formulations of the network)	13 (68.4%)	13 (68.4%)	
20	Results, individual studies	Present simplified summary data for each intervention group, effect estimates and confidence intervals; modified approaches may be used in larger networks	19 (100%)	19 (100%)	
21	Results, synthesis of results	Present the results of each meta-analysis performed including confidence/credible intervals	19 (100%)	19 (100%)	
23	Results, additional analyses	Provide the results of other analyses (e.g., sensitivity/subgroup analyses, meta-regression, alternative geometries, alternative prior distributions)	11 (57.9%)	11 (57.9%)	
25	Discussion, limitations	Discuss the limitations and comment on the validity of the assumptions such as transitivity and consistency, comment on concerns of network geometry	19 (100%)	18 (94.7%)	
	**Items included exclusively in PRISMA-NMA**		PRISMA n (%)	PRISMA-NMA n (%)	
				N = 19	N = 17 ^a^
S1	Methods, geometry of the network	Describe the methods used to explore the network geometry, including how the evidence was graphically summarized	-	6 (31.6%)	4 (23.5%) ^a^
S2	Methods, assessment of consistency	Describe the statistical methods used to evaluate the agreement between direct and indirect evidence	-	15 (78.9%)	13 (76.5%) ^a^
S3	Results, network structure	Provide a network graph of the included studies	-	18 (94.7%)	17 (100%) ^a^
S4	Results, summary of geometry	Provide a brief overview of the network characteristics	-	18 (94.7%)	17 (100%) ^a^
S5	Results, exploration of inconsistency	Describe results from investigations of inconsistency; this may include such information as measures of model fit to compare factors such as consistency and inconsistency models, or *p*-values	-	15 (78.9%)	13 (76.5%) ^a^

PRISMA checklist items were summarized. To see complete statements and checklists, access [http://www.prisma-statement.org, accessed on 15 July 2022]. PICOS, Participants, Interventions, Control group, Outcomes, Study design; SUCRA, Surface Under the Cumulative Ranking Curve. ^a^ Considered only articles published after the PRISMA-NMA publication in June 2015 (*n* = 17).

**Table 4 healthcare-10-02371-t004:** Checklist of the reporting standard guideline assessment of the included NMAs.

Common Items for PRISMA and PRISMA-NMA																					Summary		
No.	Items	1	2	3	4	5	6	7	8	9	10	11	12	13	14	15	16	17	18	19	C	P	N
4	Introduction, objectives	1	1	1	1	1	1	1	0	1	1	1	1	1	1	1	1	1	1	1	18	0	1
5	Methods, protocol and registration	1	1	1	1	0	0	1	0	1	1	1	1	1	1	1	0	1	0	0	13	0	6
7	Methods, information sources	1	1	1	1	1	1	1	1	1	1	1	1	1	1	1	1	1	1	1	19	0	0
8	Methods, search	1	1	1	1	1	1	1	1	1	1	1	1	1	1	1	1	1	1	1	19	0	0
9	Methods, study selection	1	1	1	1	1	1	1	1	1	1	1	1	1	1	1	0	1	1	1	18	0	1
10	Methods, data collection process	1	1	1	1	1	1	1	1	1	1	1	1	1	1	1	1	1	1	1	19	0	0
11	Methods, data items	1	1	1	1	1	0	1	0	0	0	1	1	1	1	0	0	1	1	1	13	0	6
12	Methods, risk of bias within studies	1	1	1	1	1	1	1	0.5	1	1	1	1	1	1	1	1	1	1	1	18	1	0
15	Methods, risk of bias across studies	0	0	1	1	1	0	1	1	1	0	1	1	0	1	1	0	1	1	1	13	0	6
17	Results, study selection	1	1	1	1	1	1	1	1	1	1	1	1	1	1	1	1	1	1	1	19	0	0
18	Results, study characteristics	1	1	1	1	1	1	1	1	1	1	1	1	1	1	1	1	1	1	1	19	0	0
19	Results, risk of bias within studies	1	1	1	1	1	1	1	1	1	1	1	1	1	1	1	1	1	1	1	19	0	0
22	Results, risk of bias across studies	0.5	0.5	1	1	1	0.5	1	1	0.5	0.5	1	1	0.5	1	1	0.5	1	1	1	12	7	0
24	Discussion, summary evidence	1	1	1	1	1	1	1	1	1	1	1	1	1	1	1	1	1	1	1	19	0	0
26	Discussion, conclusions	1	1	1	1	1	1	1	1	1	1	1	1	1	1	1	1	1	1	1	19	0	0
27	Funding	1	1	0	0	0	0	0	1	1	0	0	0	0	0	1	0	1	0	0	6	0	13
PRISMA items modified for PRISMA-NMA																					Summary		
No.	Section	1	2	3	4	5	6	7	8	9	10	11	12	13	14	15	16	17	18	19	C	P	N
1	Title	1	1	1	1	1	1	1	1	0.5	1	1	1	1	1	1	1	1	0.5	1	17	2	0
2	Abstract, structured summary	0.5	0.5	0.5	0.5	0.5	1	0.5	0.5	0.5	0.5	0.5	0.5	0.5	0.5	0.5	0.5	1	0.5	0.5	2	17	0
3	Introduction, rationale	1	1	1	1	1	1	1	1	1	1	1	1	1	1	1	1	1	1	1	19	0	0
6	Methods, eligibility Criteria	0.5	0.5	0.5	0.5	0	0.5	0.5	0.5	0	0.5	0	0.5	0	0	0.5	0	1	0	0	1	10	8
13	Methods, summary measures	1	1	1	0	1	1	0	1	1	1	1	1	1	1	1	0	1	1	1	16	0	3
14	Methods, planned analyses	1	1	0.5	0.5	0.5	0.5	0.5	1	1	0.5	1	1	0.5	0.5	1	1	0.5	1	0.5	9	10	0
16	Methods, additional analyses	0	1	0	0	1	1	1	1	1	1	1	1	0	1	1	0	0	1	1	13	0	6
20	Results, individual studies	1	1	1	1	1	1	1	1	1	1	1	1	1	1	1	1	1	1	1	19	0	0
21	Results, synthesis of results	1	1	1	1	1	1	1	1	1	1	1	1	1	1	1	1	1	1	1	19	0	0
23	Results, additional analyses	0	1	0	0	1	0	1	1	1	1	1	1	0	1	1	0	0	1	0	11	0	8
25	Discussion, limitations	1	1	1	1	1	0.5	1	1	1	1	1	1	1	1	1	1	1	1	1	18	1	0
Items included exclusively in PRISMA-NMA																					Summary		
No.	Section	1	2	3	4	5	6	7	8	9	10	11	12	13	14	15	16	17	18	19	C	P	N
S1	Methods, geometry of the network	0	0	1	0	0	0	1	0	0	1	1	1	0	0	0	0	1	0	0	6	0	13
S2	Methods, assessment of consistency	1	1	1	0	1	1	0	1	0	1	1	1	0	1	1	1	1	1	1	15	0	4
S3	Results, network structure	1	1	1	1	1	1	1	1	1	1	1	0	1	1	1	1	1	1	1	18	0	1
S4	Results, summary of network geometry	1	1	1	1	1	1	1	1	1	1	1	0	1	1	1	1	1	1	1	18	0	1
S5	Results, exploration of inconsistencies	1	1	1	0	1	1	0	1	0	1	1	1	0	1	1	1	1	1	1	15	0	4
	Summary	26.5	28.5	27.5	23.5	27	24	26.5	26.5	25.5	27	29.5	28	22.5	28	29	21	29.5	27	26			
	Quality assessment	H	H	H	M	H	M	H	H	M	H	H	H	M	H	H	M	H	H	H			

C, Completely reported; P, Partially reported; N, Not reported; H, High; M, Moderate.

**Table 5 healthcare-10-02371-t005:** Descriptive statistics of the NMAs included in this evaluation.

	Quality Score	Year	Impact Factor	Study Number	Participants
Mean	26.5	2018	5.20	42.2	3406
Median	27.0	2018	3.61	19.0	1254
Standard deviation	2.32	2.17	3.62	55.6	5533
Interquartile range	2.25	1.50	4.53	18.5	1884
Range	8.50	7.00	10.5	239	23,394
Minimum	21.0	2013	1.55	6.00	263
Maximum	29.5	2020	12.0	245	23,657
25th percentile	25.8	2018	2.59	16.5	834
50th percentile	27.0	2018	3.61	19.0	1254
75th percentile	28.0	2019	7.12	35.0	2718

**Table 6 healthcare-10-02371-t006:** Descriptive statistics of the reporting standard guidelines score.

	Group	N	Mean	Median	SD	SE
Quality score	High	9	27.5	27.0	1.46	0.486
	Low	10	25.6	26.3	2.62	0.831

SD, Standard deviation; SE, Standard error.

## Data Availability

All data relevant to the study are included in the article. Data were collected from studies published online or publicly available, and specific details related to the data will be made available upon request.

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
