# Peer review of "Evaluation of the Reporting Standard Guidelines of Network Meta-Analyses in Physical Therapy: A Systematic Review"

_healthcare, 2022, doi:10.3390/healthcare10122371_

Round 1
Reviewer 1 Report
The proposed systematic review paper regarding an evaluation of the methodological quality of network meta-analyses in physical therapy is significant as the concept of network meta-analysis (NMA) has not been thoroughly evaluated in terms of quality.
The paper has the required format and structure of a review paper. It has a summarized introduction regarding the research area involving network meta-analysis and methodological quality in physical therapy. The materials and methods section describe the comprehensively screened eligible studies from well-established databases. The authors have proposed a detailed search strategies for each database to identify the number of records within the database. As for the methodological and reporting quality assessment tools, the authors made use of PRISMA and PRISMA-NMA checklists to identify the relevant information.
Scientific soundness is a fundamental ethical requirement of all research is that it is scientifically sound. The proposed works has the required scientifically soundness as it proposes an evaluation of the methodological quality of network meta-analysis using quality assessment tools.
The paper integrates a total of 29 references, out of which only 13 are either studies or research papers published during the last 5 years.
Sources used for review papers require more “cutting edge” research, as the research field of healthcare is changing quickly with the acquisition of new knowledge. I consider that within this field, sources published in the past 2-3 years is mandatory as they provide a good benchmark and reflect the newest discoveries, processes, and best practices. The paper references only a total of 4 papers/studies which have been published in the past 3 years.
My only concern is the following:
A review paper should reference more recently published papers/studies.
Author Response
We attached the file.

Reviewer 2 Report
The authors proposed a very interesting systematic review on Network Meta Analyses focused on Physical Therapy. The paper presents in detail the method of gathering NMAs-based publications, the selection process and criteria, data collection, standards of comparison, and final reporting.
Although a very interesting paper, I ended up with a sensation that more productive outcomes could have been achieved, based on the effort shown by the authors in undertaking this research.
Addressing my concerns further, I would the authors take in consideration the following:
· The literature review is a bit unorganised as single paragraphs discuss more that one idea and some of the ideas are repeated. Please have a look to this.
· The statistical analysis of Table 7 is unclear (not properly explained), it seems out of context and, perhaps, unnecessary as I cannot see how it adds value to support paper’s conclusion.
· How the authors specifically address (and conclude) the main purpose of this research stated in line 66, especially when the concept of “quality” has not been properly defined (Quality Score is shown in tables 5, 6 and 7. But its definition is never introduced)?
· Following my previous point, it seems that the authors focused on the appropriate criteria and standards needed to perform NMAs. However, the appropriateness of the research is not assessed when following the standards correlate with the conclusions achieved in the papers. In other words, it may be happening that the NMAs agreed in the results and conclusions achieved with the treatments despite the fact of not meeting the recommended standards.
Thanks.
Author Response
We attached the file.

Reviewer 3 Report
Thank you for giving me this opportunity to review this article. The article is well written, though I have some minor concerns regarding the article.
The title of the study should be self-explanatory.
Use the mesh keywords.
Abstract:
1. Include a brief background of the review.
2. Include the character of the study articles.
3. Mention the outcome measures measured for the review.
Manuscript
1. Fix with the term physiotherapist or physical therapist throughout the review.
2. Mention the clinical significance of this review to clinicians, researchers, and patients.
3. Searching strategy – physical medicine journals or physical therapy journals?
4. The selection criteria are not specific – include exclusion criteria.
5. Mention the future recommendations of the study.
Author Response
We attached the file.

Reviewer 4 Report
This is an interesting study, of relevance to the network meta-analysis in physical therapy.
I have some comments and recommendations for revision/further clarification.
Introduction
- - Lines 60-67 (Page 2): These sentences are long and the aim could be not clear. Please, review these sentences for a better understanding and clarifying the aim of the study.
Materials and Methods
- - Lines 82-84 (Page 2): Please review this sentence for a better understanding.
- -Lines 93-99 (Page 3): Please review keywords and search strategies. Some terms are repeated and some words are considered as free text (txt word) and MeSH terms at the same time. Additionally, Table 2 should be reviewed, since the same situation are repeated in this table. Search strategies should be consistent with keywords and repetition of terms should not be included.
- - Lines 169-170 (Page 5): Perhaps you could consider mentioning a brief description related to statistical analysis for a better understanding.
Discussion
- Limitations and implications of the present study should be included.
Finally, there are some issues related to English language. Please, review English language throughout the manuscript.
Author Response
We attached the file.

Round 2
Reviewer 2 Report
The authors kindly have introduced all the suggestions and properly answered all the question relevant to their research. Thus, no further observations.
Reviewer 4 Report
The authors did a great job responding to my suggestions.